# Effects of plant diversity on productivity strengthen over time due to trait-dependent shifts in species overyielding

Plant diversity effects on community productivity often increase over time. Whether the strengthening of diversity effects is caused by temporal shifts in species-level overyielding (i.e., higher species-level productivity in diverse communities compared with monocultures) remains unclear. Here, using data from 65 grassland and forest biodiversity experiments, we show that the temporal strength of diversity effects at the community scale is underpinned by temporal changes in the species that yield. These temporal trends of species-level overyielding are shaped by plant ecological strategies, which can be quantitatively delimited by functional traits. In grasslands, the temporal strengthening of biodiversity effects on community productivity was associated with increasing biomass overyielding of resource-conservative species increasing over time, and with overyielding of species characterized by fast resource acquisition either decreasing or increasing. In forests, temporal trends in species overyielding differ when considering above- versus below-ground resource acquisition strategies. Overyielding in stem growth decreased for species with high light capture capacity but increased for those with high soil resource acquisition capacity. Our results imply that a diversity of species with different, and potentially complementary, ecological strategies is beneficial for maintaining community productivity over time in both grassland and forest ecosystems.

Plant diversity often increases community productivity in natural and experimental ecosystems[1–4]. Higher productivity in diverse plant communities compared with the average of component species in monocultures (which we define herein as community overyielding) can arise through niche differentiation among species (complementarity effects) and/or dominance of highly productive species (positive selection effects)[5]. Community overyielding (or underyielding) is the net effect of species-level responses to plant diversity due to species interactions. Individual species may have higher or lower productivity in diverse communities than in their monocultures, leading to species-level overyielding or underyielding respectively[6]. Evidence suggests that community overyielding in mixtures occurs when strong overyielding of a few species overcompensates for underyielding and/or

neutral responses in others[6,7], or when most or all species overyield (neither requires all species to contribute equally[5,8,9]). Yet, to what extent changes in species-level contributions (e.g., species overyielding) shape temporal dynamics of plant diversity effects on community productivity remains poorly understood[10].

Understanding the temporal dynamics of plant diversity-productivity relationships can increase our ability to predict the long-term effects of biodiversity change on ecosystem functioning[11–13], particularly when considering the accelerating loss of species or introductions of exotic species caused by human activities[8,14,15]. Positive biodiversity-productivity relationships in grassland and forest ecosystems often strengthen over time[11,13,16–19]. This temporal strengthening of biodiversity effects on productivity may result from

✉e-mail: litingz@umich.edu; eryan@des.ecnu.edu.cn

an increase in productivity in diverse plant communities, a decrease in productivity in monocultures, or both[13,20,21]. Such temporal responses are likely driven by shifts in plant-plant interactions[22], resource acquisition and use[18], plant-soil feedback effects that increase soil fertility[23,24], and biotic interactions with other trophic levels[21,25]. For example, the decreasing performance of plant monocultures over time might be the result of an accumulation of species-specific soil-borne pathogens[21,25], while the increasing performance of more diverse communities could be due to increasing complementary resource use among species[11,13,16,17]. The magnitude of these shifts may depend on plant ecological strategies that influence species performance and interactions between species, and hence, ecosystem functions and processes[10,18,26–28].

One way in which plant diversity can increase the performance of communities over time is through a shift in the relative abundance of species representing different ecological strategies[10,29]. Some of these strategies can be represented by above- and belowground functional traits[30,31]. One important gradient in these traits is the resource economics spectrum[30,32], which ranges from acquisitive to conservative plant strategies in terms of resource uptake and tissue turnover. Specifically, plant species with acquisitive strategies are expected to have fast rates of resource uptake as well as short-lived leaves or roots, with high specific leaf area (SLA), leaf nitrogen content (LNC), specific root length (SRL), and root nitrogen content (RNC), but low leaf dry matter content (LDMC) and root tissue density (RTD). Alternatively, plant species with conservative strategies are slow in resource acquisition, but able to survive and thrive in low-resource conditions[30–32]. In mixed-species communities, acquisitive and conservative species may be more or less likely to over- or underyield due to their differing abilities in resource preemption and stress tolerance[33]. For example, overyielding species in young mixed-species communities are usually adept at acquiring large amounts of nutrients per plant, or are N-fixers (i.e., legumes)[6,29], or intercept more light than underyielding species[34,35]. Yet, resource availability at the system scale, per capita, or both, changes during the development of the plant community[36,37], likely shifting the species-level contributions to community overyielding over time. For instance, the dominant species in mixed-species communities at early stages were gradually replaced by slower-growing, deep-rooting species in both grasslands and forests[29,38]. These shifts are likely related to differences in plant strategies defined through functional traits (e.g., differences in plant nutrients content[23,24]).

At the early stages of plant community development, above-ground space and soil volume are largely unoccupied, resources (i.e., light, water, and nutrients) are typically abundant (but their availability might be context-dependent), and competition among individuals for resources is minimal. Unoccupied above- and belowground spaces with abundant resources allow acquisitive species to grow faster than conservative species in mixed-species communities[35,39,40]. Once individuals begin interacting, acquisitive species will initially benefit more from competing with conservative species[18]. It is possible that over-yielding in acquisitive species overcompensates for underyielding of conservative species that have low competitive abilities[10,41]. The greater overyielding in acquisitive species is also possible when resource partitioning in space or time among constituent species enhances the capture of key resources and growth by acquisitive species[42].

During plant community development, niche differentiation (e.g., via resource partitioning) among plant species is expected to promote greater community-level uptake of limiting resources in diverse plant communities[11,13,16,17]. However, resource availability per individual often decreases through time[36]. For example, increasing leaf area in diverse plant communities over time reduces light availability to each individual leaf or plant on average (i.e., greater light interception at the top of the canopy leads to less light transmission to leaves in the lower strata of the same individual's crown and those of understory plants[43]). Similarly, increased above- and belowground plant growth causes greater water demand but lower water availability per unit root area[36]. Although increases in nitrogen mineralization, driven by plant diversity, can partially compensate for higher nitrogen demand, nitrogen availability per unit root mass/volume often declines[36]. Furthermore, management activities (e.g., repeated biomass harvest or litter removal) can also cause soil nutrient availability to decrease during community development[29,36]. Decreases in resource availability from any or all of the above sources over time in mixed-species communities may increase overyielding in conservative species which exhibit greater advantage for resource acquisition than acquisitive species in resource-limited environments[29], and which often exhibit higher resource use efficiency (create more biomass per unit of resource consumed). Overyielding of acquisitive species may either decrease over time due to their lower capacity to acquire resources as environments become more resource-limited (due to competition in mixtures), or increase slightly due to the decreasing performance of their plant monocultures (e.g., caused by accumulated soil-borne pathogens or intense intraspecific competition[13,21]). The increased overyielding of conservative species over time as communities develop may either partially, fully or overcompensate for a decrease in species overyielding of acquisitive species, or lead to community overyielding in combination with increased overyielding of acquisitive species.

Here, we investigate if the temporal strengthening of plant diversity effects on aboveground productivity (Fig. 1A) is related to shifts in contributions of acquisitive and conservative species to increased productivity in diverse plant communities over time (Fig. 1B, C). This shift may happen simultaneously with other biological mechanisms through which diversity effects act (e.g., resource partitioning[42,44], interspecific competition leading to the dominance of species with particular traits[29], self-thinning[38], feedback on soil fertility[23], or other biotic interactions[21]). Specifically, we hypothesize that acquisitive plant species contribute more to overyielding in mixed-species communities than conservative species during the early stages of community development (H1), whereas conservative species contribute more to overyielding than acquisitive species at later stages (H2). Furthermore, we predict that the timing of shifts in contributions to community overyielding from acquisitive to conservative species differs between grassland and forest experimental ecosystems (H3). Community dynamics in forests are relatively slow compared to those in grasslands due to the differences in plant physiology and structure, and longer lifespan of woody species[45,46]. In addition, the ways in which the species overyielding shifts differ between grasslands (mainly at the population level, i.e., relatively fast changes in the number of individuals rather than size[47]) and forests (mainly at the individual level, i.e., relatively slow processes of tree growth, canopy space filling, and mortality[38]). We, therefore, expect that temporal shifts in overyielding from acquisitive to conservative species in forests may be slower than those in grasslands.

We tested these hypotheses using temporal data from 65 biodiversity experiments across the globe that manipulated plant species richness in grassland (39) and forest (26) communities (Supplementary Table 1 and Supplementary Fig. 1) including 166 herbaceous and 134 tree species, respectively. To elucidate temporal changes on diversity effects, we quantified community overyielding, complementarity and selection effects. To capture conservative and acquisitive resource-acquisition strategies of plant species, we selected plant economic traits including SLA, LNC, LDMC, SRL, RNC, and RTD. Trait data were mainly obtained from the Plant Trait Database (TRY[48]), the Global Root Trait (GRooT) database[49], and site-specific measurements from certain experiments. We find that the temporal strength of diversity effects is determined by changes in species

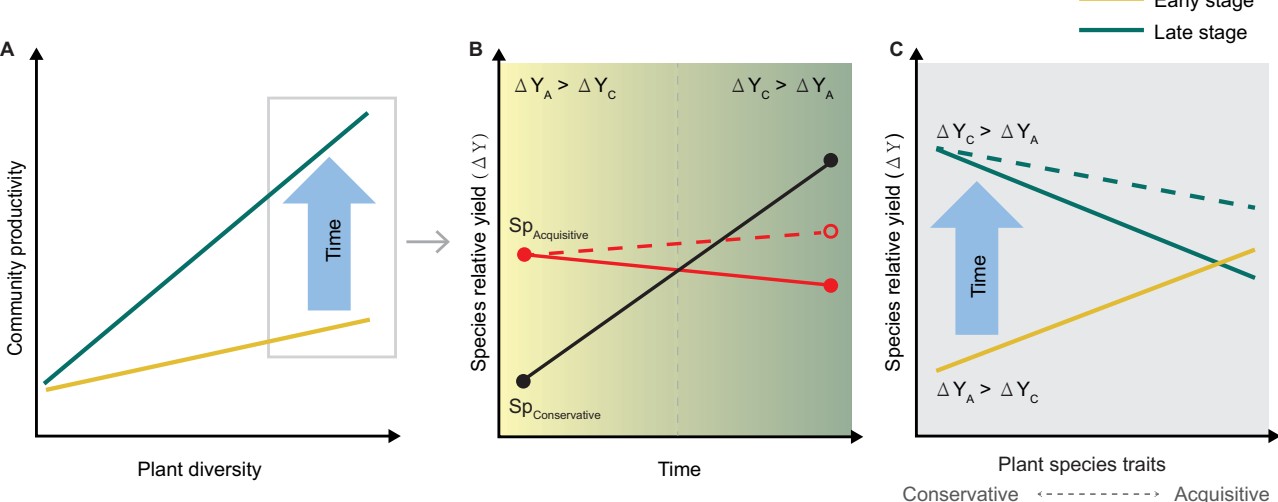

**Fig. 1 | Overview of how temporal changes of plant diversity effects on community productivity are expected to be related to trait-dependent shifts in species-level overyielding. A** Effects of plant diversity strengthen over time as productivity increases more quickly in diverse plant communities. **B** At early stages, species-level overyielding in diverse plant communities is higher for acquisitive ($\Delta Y_A$) than for conservative ($\Delta Y_C$) species. Over time, the overyielding of conservative species ($SP_{Conservative}$) is expected to increase, whereas the overyielding of acquisitive species ($SP_{Acquisitive}$) is expected either to decrease (solid red line) or increase slightly due to the decreasing performance of monocultures (dotted red line). **C** Contributions to community overyielding shift over time, from greater overyielding for acquisitive species at early stages to greater overyielding for conservative species at later stages of community development, regardless of whether overyielding of acquisitive species decreases (solid green line) or slightly increases (dashed green line).

overyielding, with temporal trends of species-level overyielding shaped by plant ecological strategies. However, these patterns differ between grassland and forest experimental systems. In grasslands, temporal strengthening was associated with increases in the overyielding of resource-conservative species and either decreasing or increasing for fast resource species. In forests, overyielding in stem growth decreased for species with high light capture capacity but increased for those with high soil resource acquisition capacity.

## Results

### Community-level overyielding strengthens over time
The effect of plant species richness on community-level productivity and overyielding (i.e., species mixture effects on community productivity calculated as community log response ratio) increased significantly over time in both grasslands and forests (Fig. 2). In grasslands (Fig. 2A, B), this was primarily due to increases in the complementarity effect, whereas the selection effect decreased over time (Supplementary Table 2 and Supplementary Fig. 2A, B). In forests (Fig. 2C, D), the increase in the slope of the species richness-basal area increment over time was not related to changes in either complementarity effects or selection effects (Supplementary Table 2 and Supplementary Fig. 2C, D). When considering the accumulated basal area in forests, the strengthened positive plant species richness effect was driven by temporal increases in both complementarity and selection effects (Fig. 2E, F, Supplementary Table 2 and Supplementary Fig. 2E, F).

### Species contribute differently to community overyielding
In both grasslands and forests, species contributions to community overyielding varied widely across all experimental years (Fig. 3). Yet, in most cases, community overyielding was attributable either to overyielding of about half of the species while the remaining species were underyielding, or to all of the constituent species overyielding (Fig. 3A). On average, more than half of the mixed-species communities exhibited overyielding across all experimental years. Across all overyielding communities, the highest overyielding species in a community contributed more than half of proportional community-level overyielding, regardless of the number of species that overyielded in a community (Fig. 3B).

### Trait-dependent species-level overyielding changes over time
In grasslands, species-level biomass overyielding (calculated as species log-response ratio) on average increased with plant species richness, and its effects strengthened over a time of ten years (Table 1 and Supplementary Fig. 3A). In the early years of grassland community development, acquisitive species with higher SLA and SRL and lower RTD (captured by the first principal component of herbaceous species trait space; Supplementary Table 3 and Supplementary Fig. 4A), and higher LNC (captured by the second principal component) had higher biomass overyielding in mixed-species communities than conservative species with the opposite trait values (Fig. 4A, B, Supplementary Fig. 8A, B, D, E and Supplementary Table 5). Acquisitive species either decreased or increased biomass overyielding over time, whereas overyielding of conservative species constantly increased over time, and their overall increase exceeded that of acquisitive species (Supplementary Fig. 7A, B). The biomass overyielding of species with higher RTD and lower LNC outpaced that of acquisitive species (i.e., lower RTD and higher LNC) in later years (Supplementary Fig. 8D, E). This temporal shift was also observed when we included all of the data across 18 years (only three experiments > 10 years; Supplementary Fig. 9).

There was an imbalance in the lengths of the grassland experiments, i.e., many grassland experiments collected data for less than four years (Supplementary Table 1). Therefore, we conducted further analyses to test whether the trait-dependent temporal shifts in species overyielding were consistent among the grassland experiments with more than four years of data versus all grassland experiments (Methods). We found that the temporal trends of the restricted longer-term dataset (i.e., only 6 experiments) yielded similar results (Supplementary Figs. 12 and 15) as those presented above using the full datasets (i.e., all 39 experiments). In the restricted longer-term dataset, the overyielding of conservative species with lower SLA, SRL and LNC constantly increased over time, although the slopes changed for certain traits (e.g., Supplementary Fig. 15C–E) due to differences in the number and identity of species.

In forests, there were no positive effects of increasing tree species richness on average species-level overyielding (based on either annual basal area increment or accumulated basal area), and no evidence that

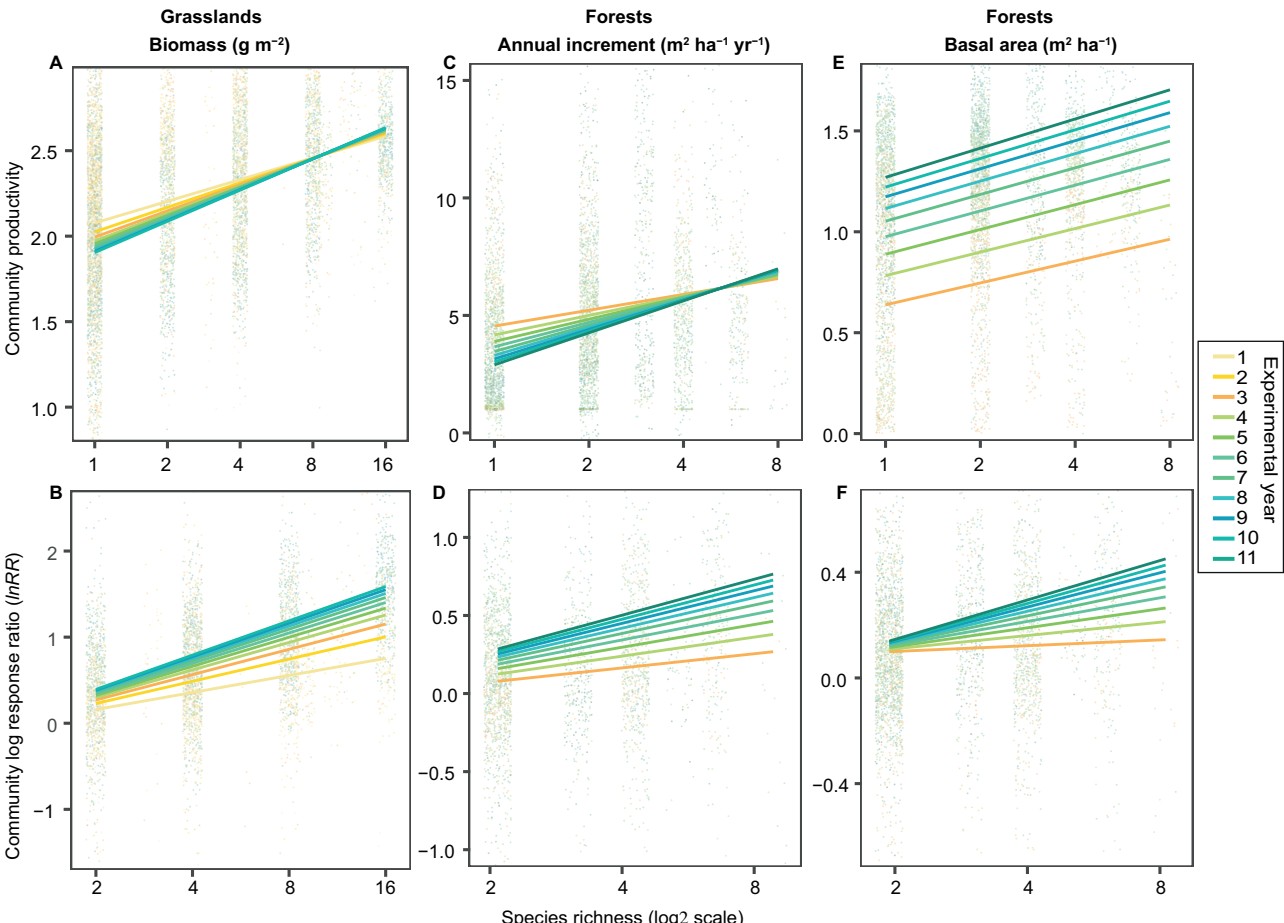

**Fig. 2 | Effects of plant species richness on community productivity and community overyielding over time. A, B** Effects on community productivity and community overyielding (log response ratio (*lnRR*) of productivity of a mixture divided by the mean productivity of all monocultures of the component species) in terms of total aboveground biomass (log-10 scale) in grasslands (*n* = 39). Effects on (**C, D**) annual basal area increment and (**E, F**) total accumulated basal area (log-10 scale) in forests (*n* = 26). Points are community-level values for each plot in the respective year. Lines are mixed effect model fits across all experiments. For grasslands, species richness ($F_{1,1317}$ = 157.8, $P$ < 0.001), year ($F_{1,5976}$ = 21.4, $P$ < 0.001), and the species richness × year interaction ($F_{1,5976}$ = 8.0, $P$ = 0.005) significantly affected aboveground biomass, while species richness ($F_{1,716}$ = 78.1, $P$ < 0.001) and the species richness × year interaction ($F_{1,3482}$ = 40.0, $P$ < 0.001) significantly

affected community overyielding. In forests, annual basal area increment was significantly affected by year ($F_{1,3474}$ = 31.9, $P$ < 0.001), and the species richness × year interaction ($F_{1,3474}$ = 7.6, $P$ = 0.006), while community overyielding was significantly affected by the species richness × year interaction ($F_{1,1934}$ = 12.6, $P$ < 0.001). Forest accumulated basal area was significantly affected by species richness ($F_{1,1348}$ = 13.6, $P$ < 0.001), year ($F_{1,4023}$ = 2229, $P$ < 0.001) and the species richness × year interaction ($F_{1,4023}$ = 12.6, $P$ < 0.001), while community overyielding was significantly affected by species richness ($F_{1,781}$ = 4.9, $P$ = 0.027), and the species richness × year interaction ($F_{1,2611}$ = 19.4, $P$ < 0.001). Reported $P$ values were calculated from one-sided $F$-tests. Refer to Supplementary Table 2 for more details. Y-axis was trimmed to enhance resolution comparing model fit lines (5% extreme values are not visible).

the effects of species richness on species-level overyielding increased over a time of eleven years (neither species richness nor species richness × year interaction was significant; Table 1). From the early to the later years, in most mixed-species communities across tree species richness levels, half of the species overyielded and the other species underyielded (Fig. 3A and Supplementary Fig. 3B, C). In the early years of mixed-species community development, acquisitive tree species with higher LNC and RNC (captured by the first principal component of tree species trait space; Supplementary Table 3 and Supplementary Fig. 4C), higher SRL, and lower RTD (represented by the second principal component) exhibited higher overyielding than conservative species with the opposite trait values (Fig. 4C, D, Supplementary Fig. 10 and Table 6). Over time, species-level overyielding of annual basal area increment for species with higher SLA and LNC decreased, while that for species with lower SLA and LNC increased (Fig. 4C and Supplementary Figs. 10A, B). In contrast, the overyielding in annual basal area increment for acquisitive tree species with higher RNC and SRL and lower LDMC and RTD increased over time (Fig. 4D and Supplementary Fig. 10C–F). In terms of accumulated basal area, species-level

overyielding was greater for acquisitive species with higher LNC, RNC, and SRL and lower RTD than for conservative species with the opposite trait values in the early years of tree community development, and strengthened over time (Fig. 4E, F and Supplementary Fig. 11B–D, F).

## Discussion

We found that species overyielding in mixed-species plant communities depends on species-specific traits and shifts over time in both grassland and forest experimental ecosystems. In grasslands, the species that contributed most to increased productivity in diverse plant communities (i.e., greater extent of community-level overyielding) shifted over time from acquisitive to conservative. In contrast, tree species with root trait values associated with fast soil resource acquisition drove an increase in community-level overyielding in young mixed-species tree communities. This temporal trend in tree communities reflects the shifting role of acquisitive species in contributing to enhanced productivity in diverse communities, progressing from light capture to belowground soil resource

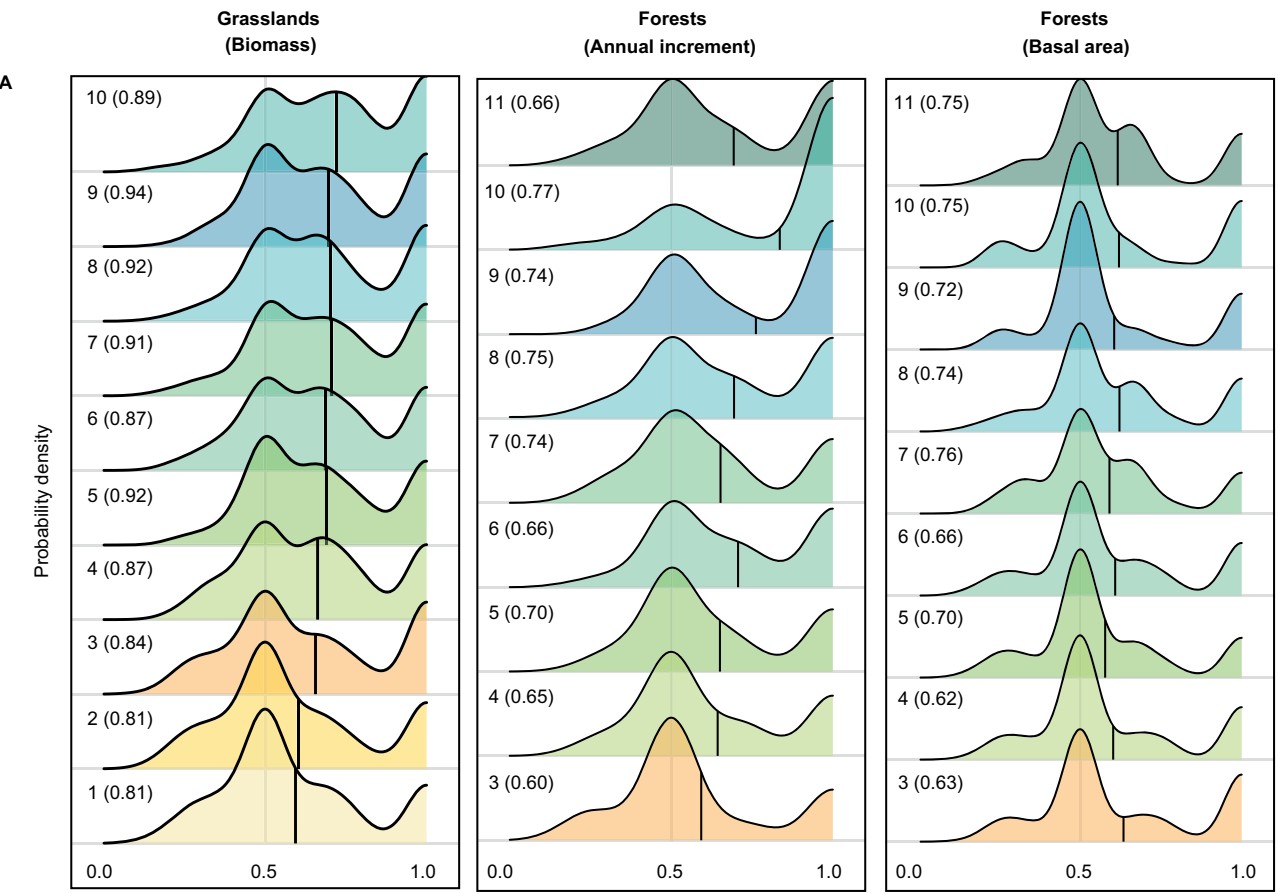

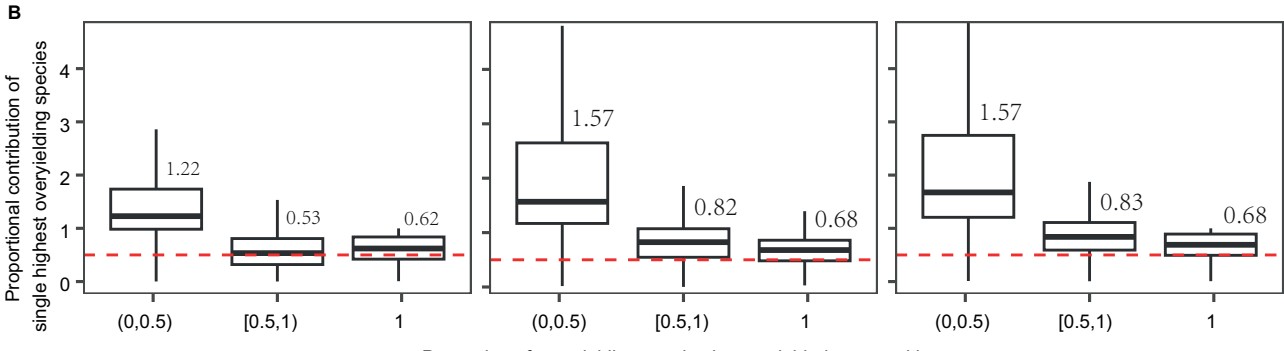

**Fig. 3 | The contribution of species overyielding to community overyielding in grasslands ($n$ = 39) and forests ($n$ = 26) experimental ecosystems. A** The probability density of the proportion of overyielding species (with a positive species *lnRR*) in overyielding communities over time. Numbers on the left sides of each panel represent experimental years and the numbers in the parentheses represent the proportion of overyielding communities in the corresponding year. The vertical lines indicate the mean values of the proportion of overyielding species in overyielding communities each year. **B** Box plots showing the proportional contribution of the single highest overyielding species to community-level overyielding ($CO_{max}$) when less than half of all species overyielded (0, 0.5), more than half of the species

overyielded (0.5, 1), or when all species overyielded (1) in overyielding communities across years. Numbers above the boxes show the median $CO_{max}$ for each group. $CO_{max}$ can be larger than 1 when overyielding in some species overcompensates for underyielding in other species, whereas $CO_{max}$ ranges between 0 and 1 when all species overyielded in overyielding communities. Note that the proportional contributions of all species to community-level overyielding add up to 1. For each boxplot, the horizontal lines inside the box represent the median, the lower and upper ends of the boxes represent the 25th and 75th percentiles, and the lower and upper whiskers extend from the hinge to the largest and lowest value, respectively.

exploitation. Thus, changes in the relative contribution of acquisitive and conservative species to community overyielding underpins the increasing strength of positive biodiversity-productivity relationships through time.

Interspecific interactions among species drive increased productivity in diverse plant communities[6,16,42,50]. Differences in species' contributions to community overyielding may be driven by

asymmetric competition between plant species in diverse communities[28,33]. Some species with specific functional trait values exhibit greater overyielding by intercepting or using disproportionate amounts of resources (i.e., resource preemption). For example, several studies show that strong competitors for nitrogen (e.g., C4 grasses) or nitrogen-fixing plant species overyield more in mixed-species grassland communities where nitrogen is the limiting soil resource[6].

**Table 1 | Linear mixed-effects models for effects of plant economics traits (represented by traits PC1 and traits PC2), species richness (SR), year, and their interactions on species overyielding (or underyielding) of aboveground biomass in grasslands and annual basal area increment and accumulated basal area in forests**

| Grasslands | | | | Forests | | | | | | | |
|---|---|---|---|---|---|---|---|---|---|---|---|
| Species log-response ratio (representing species overyieling or underyielding) | | | | | | | | | | | |
| | Biomass | | | | Annual basal area increment | | | | Accumulated basal area | | | |
| | df | ddf | *F* | *P* | df | ddf | *F* | *P* | df | ddf | *F* | *P* |
| Traits PC1 | **1** | 3536 | 1.68 | 0.195 | 1 | 2430 | 0.78 | 0.378 | 1 | 2460 | 0.64 | 0.423 |
| SR | 1 | 3536 | 2.35 | 0.125 | 1 | 2430 | 2.95 | 0.086 | 1 | 2460 | 2.61 | 0.106 |
| Year | 1 | 14717 | 0.41 | 0.520 | 1 | 6028 | 0.75 | 0.386 | 1 | 7783 | 2.17 | 0.140 |
| Traits PC1 × SR | 1 | 3536 | 0.96 | 0.327 | 1 | 2430 | 2.50 | 0.114 | 1 | 2460 | 0.01 | 0.938 |
| Traits PC1 × Year | 1 | 14717 | 9.97 | **0.002** | 1 | 6028 | 6.46 | **0.011** | 1 | 7783 | 3.98 | **0.046** |
| SR × Year | 1 | 14717 | 22.16 | **<0.001** | 1 | 6028 | 2.56 | 0.110 | 1 | 7783 | 0.19 | 0.664 |
| Traits PC2 | 1 | 3536 | 0.34 | 0.557 | 1 | 2430 | 3.74 | 0.053 | 1 | 2460 | 0.33 | 0.567 |
| SR | 1 | 3536 | 6.16 | **0.013** | 1 | 2430 | 2.23 | 0.135 | 1 | 2460 | 2.88 | 0.090 |
| Year | 1 | 14717 | 1.95 | **0.163** | 1 | 6028 | 0.62 | 0.433 | 1 | 7783 | 2.43 | 0.119 |
| Traits PC2 × SR | 1 | 3536 | 6.62 | **0.010** | 1 | 2430 | 0.19 | 0.665 | 1 | 2460 | 0.18 | 0.672 |
| Traits PC2 × Year | 1 | 14717 | 19.86 | **<0.001** | 1 | 6028 | 13.18 | **<0.001** | 1 | 7783 | 11.95 | **<0.001** |
| SR × Year | 1 | 14717 | 30.82 | **<0.001** | 1 | 6028 | 2.10 | 0.148 | 1 | 7783 | 0.27 | 0.606 |

Significant *P* (*P* < 0.05) are denoted in bold. Reported *P* values were calculated from one-sided *F*-test. Refer to Supplementary Table 4 for variance components of the random effects.
*df* numerator degrees of freedom, *ddf* denominator degrees of freedom, *F* F ratios, *P* P value of the significance test.

Furthermore, taller grasses, which may indicate a greater capacity to compete for light, tend to overyield[51,52]. In some young experimental tree communities, acquisitive shade-intolerant broad-leaved or conifer species overyield, which is likely due to reduced competition from neighboring species[35,41,50]. Notably, this trait-dependent species overyielding in mixed-species communities is not only attributable to the selection of productive species (i.e., overyielding of species in mixtures that are productive in monocultures[41]), but also occurs simultaneously with resource partitioning[42], facilitation (e.g., nitrogen fixation[6]), and biotic feedback effects (e.g., release from pathogens[21,53]).

In grasslands, our results showed that temporal increases in community overyielding were linked to temporal increases in the positive relationship between species overyielding and species richness, indicating gradually enhanced complementary resource use among a greater number of the species in mixed-species communities[11,16,17]. Notably, the temporal shifts in the contributions to community overyielding of acquisitive and conservative species also contributed to the strengthening of diversity effects over time, as predicted. Due to decreased resource availability per individual/area over time[36,43], the overyielding of acquisitive species was outpaced by conservative species that are adapted to low resource availability in older mixed-species grassland communities. In addition to resource partitioning, the temporal shift in the contributions to overyielding associated with contrasting plant ecological strategies might also be attributed to biotic feedback effects[53]. Acquisitive species might be more affected by the accumulation of pathogens over time, while conservative species may be less vulnerable to pathogens because of higher carbon allocation to defense compounds and physical barriers[21]. Furthermore, the growth of conservative species is less vulnerable to climate variability and is more stable over time[54,55], which may also contribute to the increased overyielding of conservative species over time.

However, we did not find decreases in overyielding for acquisitive species with high RNC over time in grasslands. Evidence suggests that although soil $NO_3$–N availability usually decreases during community development, plant diversity increases plant-derived N inputs to soils over time[23,37]. In addition, plant diversity can increase N mineralization rate and $NH_4$–N concentration over time and decrease N leaching due to increased soil microbial biomass and higher N-uptake (with higher fine root biomass)[37,56]. These processes may compensate for the initial depletion of soil $NO_3$–N availability in mixed-species communities[36], and contribute to the N uptake of species with high N demand in later years[29].

In forests, our results showed that community overyielding increases with species richness over time, and this increase is associated with increases in species overyielding that overcompensates for underyielding in others. For example, shade-intolerant acquisitive species overyielded (in terms of mean annual basal area increment) at the expense of conservative species during the early years of the experiments. Viewed through the lens of competition for light, acquisitive species with a greater ability to capture light and higher photosynthetic capacity tend to develop their crowns rapidly in mixed-species communities to dominate the community and drive aboveground community overyielding[35,41,57]. As forest communities develop, overyielding in stem growth for species with a greater capacity to capture light may decrease over time, which may be related to their low tolerance to competition for light, combined with the progressive expansion of tree crowns and canopy closure[34,58]. However, overyielding of annual basal area increment for tree species with higher RNC and lower RTD increased over time in our study, indicating that the greater contributors to community overyielding eventually shift from light-harvesting to N-exploitative tree species. This underscores the important role of soil exploitation strategies in determining tree diversity effects on productivity during forest community development[57,59].

Although we explored the temporal shift in species overyielding in biodiversity experiments over ten years, tree communities in our study are still young compared to the lifespan of individual trees[45]. In our study, tree species with resource acquisition strategies had increasing overyielding in the accumulated basal area over time, i.e., acquisitive tree species still dominated in young tree plantations, which is consistent with previous studies[18,22,40,57]. The greater overyielding of acquisitive tree species may have larger effects as the canopy of young stands develop (~20 years[60]). We expect that the shift from greater overyielding of acquisitive to conservative species over time may occur at a longer temporal scale, e.g., when competition for available soil resources intensifies or the forest enters the stem

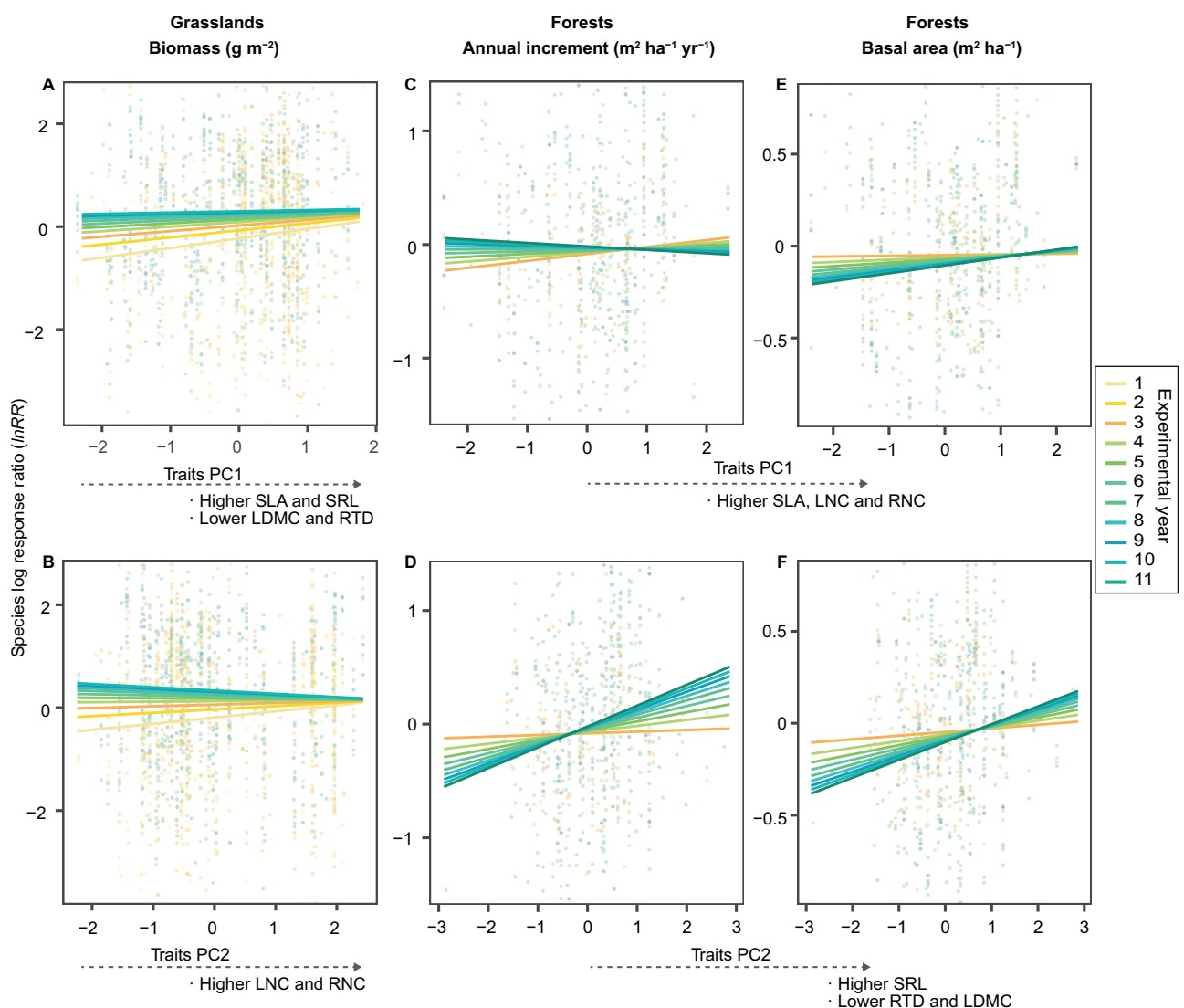

**Fig. 4 | Species overyielding (or underyielding) in mixed-species communities in relation to plant economics traits across experimental years in grassland and forest experimental ecosystems. A, B** The relationship between species log response ratio (*lnRR*; positive value indicates overyielding and negative value indicates underyielding) of aboveground biomass with plant economics traits (A: Traits PC1; B: Traits PC2) in grasslands (*n* = 39). **C-F** The relationship between species *lnRR* of annual basal area increment with plant economics traits (**C, E** Traits PC1; **D, F** Traits PC2), as well as the relationship between species *lnRR* of accumulated basal area in forests (*n* = 26). Lines are mixed-effects model fits across experiments (refer to Supplementary Figs. 5-6 for model fits within each experiment). Points represent the *lnRR* for each species and year in each experiment. Y-axis were trimmed to enhance resolution comparing model fit lines (5% extreme values are not visible). Refer to Table 1 and Supplementary Table 4 for detailed statistical analyses.

exclusion (i.e., when intense competition results in mortality) and recruitment phases of forest stand development[59,61]. However, several more decades of data may be needed to uncover the full dynamics of overyielding in forest diversity experiments. Several studies have shown that tree diversity promotes tree growth and reduces mortality[38], or has positive effects on growth but no effect on mortality[62], or even leads to higher tree growth and higher mortality[63]. Further studies using longer-term inventory data to explore the mechanisms of how tree diversity influences long-term processes (i.e., growth, mortality, and recruitment) during community development might be necessary. In addition, biotic factors such as initial plant density (e.g., higher planting density may accelerate ecological processes) and abiotic (e.g., soil characteristics, shading) factors may influence the trajectories of shifts in species overyielding and temporal dynamics of tree diversity effects on productivity[13,60]. Studies with longer temporal scales are needed to further elucidate the context-

dependency of temporal shifts in the contribution of species to community overyielding during forest community development.

We did not examine trait plasticity in response to plant diversity over time due to the lack of individual-level trait measurements across and within experiments. The incorporation of trait plasticity may improve predictive power in explaining plant diversity-ecosystem functioning relationships[34,64]. Several studies have shown similar patterns of trait variation within species in response to plant diversity, which indicates that plants likely adapt strategies to optimize resource acquisition or resource use efficiency[34,61,64]. For example, legumes have higher SLA in more diverse grassland communities, which increases light acquisition[64]. Tree individuals tend to enhance light interception in mixed-species communities through crown shape plasticity[65] and increase soil resource acquisition through root plasticity (higher RNC and lower RTD)[59]. Here, we focus on the differences between species in their relative contributions to community overyielding within

mixed-species communities (i.e., interspecific changes in trait-dependent overyielding over time at the species level). However, intraspecific trait variation may provide further insights into how the interception and acquisition of individual resources (e.g., light, water, N) determines temporal shifts in species overyielding. Future studies need more extensive trait datasets with both temporal and spatial variation to test the consequences of trait plasticity on plant diversity effects over time[17,53]. The other limitation of this study is the imbalance in dataset length (mainly in grasslands), which is a source of uncertainty in our analyses. Our results indicate that the analyses of the experiments with different lengths do not substantively alter the conclusions based on available data. More extensive synthesis with longer-term experiments would provide more critical insights into the generality of how trait-dependent shifts in species overyielding determine the temporal strength of diversity effects on ecosystem productivity in the longer term.

Overall, our results show that shifts in trait-dependent species overyielding are associated with strengthening of the positive plant diversity-ecosystem productivity relationship over time in both grassland and forest ecosystems. Considering the potential trade-offs associated with changes in resource availability over time[11,29], plant species with diverse ecological strategies not only contribute to interspecific complementary resource use within communities at a certain point in time, but also play complementary roles over time in maintaining productivity. This study extends current understanding of the temporal dynamics of diversity–productivity relationships by demonstrating how trait-based mechanisms can contribute to species selection for short- and long-term ecosystem restoration initiatives[23–25,66]. Ecosystem restoration using species mixtures with a diversity of ecological strategies is vital for both the restoration of ecosystem productivity in the short term, and the maintenance of that productivity in the long term.

## Methods

### Data acquisition and description

We assembled a database by combining data from both grassland and forest biodiversity experimental ecosystems. Experiments included in this study met the following criteria: (1) plant species richness was directly manipulated through sowing or planting and contained both mixtures and corresponding monocultures, (2) the relative abundance of species in each mixture was known, (3) aboveground plant biomass for both species- and plot-level were available, and (4) the experiment was conducted for at least three years with measurements performed in at least two different years. These experiments typically prevent plot contamination by repeatedly removing non-target species to maintain the desired composition and diversity levels. When an experiment included other treatments (e.g., fertilization), we filtered out data where other treatments existed. For grassland experiments, we used up to ten years of data per study with a range in species richness from one to 16 (if available) to reduce potential bias associated with the scarcity of experiments that lasted longer than ten years. Similarly, for forest experiments, we used up to eleven years of data with a range in species richness from one to eight (if available) and excluded data from the initial two experimental years. We excluded the first two years to avoid potential biases associated with re-planting that commonly occurred in the initial two years of some forest experiments. In total, we used data from 39 grassland and 26 forest experiments for the main analysis (see the details of the experiments included in this study in Supplementary Table 1).

### Plant functional traits

We selected six functional traits to define the resource-use strategies of the herbaceous and tree species in our dataset: specific leaf area (SLA; mm$^2$ mg$^{-1}$), leaf nitrogen content (LNC; mg g$^{-1}$), leaf dry matter content (LDMC; %), specific root length (SRL; m g$^{-1}$), root nitrogen content (RNC; mg g$^{-1}$), and root tissue density (RTD; g cm$^{-3}$). These traits are associated with the fast-slow economics spectrum and are important for explaining plant ecological strategies and community processes[30,31]. Trait values were obtained from the TRY[48], GRooT[49], the Botanical Information and Ecology Network (BIEN)[67], and AusTraits[68] databases, or from site-specific measurements when available. Even though plant trait values may differ among experimental sites, inter-specific trait variation is usually greater than intraspecific variation, and species-level trait means mainly reflect the acquisition strategies of plants as an outcome of evolutionary processes[69].

We used species mean trait values to define plant resource-use strategies, and principal component analysis (PCA) was used to quantify the difference in resource-use strategies among species. As PCA can only be performed on complete datasets, we filled the gaps in the trait datasets by imputing missing trait values[70], using the missForest R package[71]. In order to perform this process following a robust protocol, we imputed data only for species with available traits. Overall, most herbaceous species had more than five available measured traits and most tree species had more than four available measured traits (Supplementary Fig. 4 E, F). In cases where trait values could not be reliably imputed due to a lack of all trait measurements, we excluded those species from subsequent analyses (17 herbaceous species and 1 tree species). In this case, we obtained the imputed herbaceous trait dataset with a total of 166 species (94 species had complete measured traits). The imputed tree trait dataset included a total of 134 species (42 species had complete measured traits). The species-level data for SLA, LDMC, LNC, SRL, RNC, and RTD were available for 99%, 89%, 92%, 81%, 77%, and 72% of herbaceous species, respectively (see specific numbers of species with measured and imputed values for each trait in Supplementary Table 3). The data for SLA, LDMC, LNC, SRL, RNC, and RTD were available for 99%, 74%, 91%, 72%, 50%, and 60% of the tree species, respectively. In total, 85% of all trait values in the final imputation dataset for herbaceous species were gathered from empirical measurements, while the remaining 15% were imputed. For tree species, 75% of all trait values in the final imputation dataset were gathered from empirical measurements, with the remaining 25% of values imputed.

Given that evolutionarily closely related species tend to be similar in functional traits and many traits display high degrees of phylogenetic signal[70], we added phylogenetic information into the imputation algorithms. The phylogeny was obtained using the R package V.Phylomaker[72], with the GBOTB phylogeny as the backbone[73]. We then assessed the reliability of the imputation procedure[70] by using the normalized root mean square error (NRMSE) to estimate the average distance between real and artificially imputed values of species as a proportion of the range of trait values of species (Supplementary Methods 1). The NRMSE for SLA, LNC, LDMC, SRL, RNC, and RTD of tree species was 0.01, 0.04, 0.05, 0.07, 0.08, and 0.08, respectively. The NRMSE for SLA, LNC, LDMC, SRL, RNC, and RTD of tree species was 0.01, 0.06, 0.09, 0.12, 0.13, and 0.14, respectively, indicating that imputations performed in this study produced low errors and provided reliable estimates of missing trait values[74].

Using the imputed datasets, we constructed trait space for the herbaceous and tree species (Supplementary Table 3 and Supplementary Fig. 4) with PCA and extracted species scores on the first two PCA axes to represent conservative to acquisitive species. For the PCA analysis of the herbaceous species' traits, the first PC (38% of total trait variation) represents a gradient from low SLA and SRL and high LDMC and RTD to high SLA and SRL and low LDMC and RTD. The second PC (23% of total trait variation) represents a gradient from low to high LNC and RNC. For the PCA analysis of tree species' traits, the first PC (35% of total trait variation) represents a gradient from low SLA, LNC, RNC, to high SLA, LNC, and RNC. The second PC (22% of total trait variation) represents a gradient from low SRL and high RTD and LDMC to high SRL and low RTD and LDMC. For both herbaceous and tree species,

low values on both PCA axes represent the conservative strategy, and high values represent the acquisitive strategy.

## Productivity

We assembled a database by combining original data on the aboveground productivity of each species in every plot from all experiments. For grasslands, we used annual peak aboveground live biomass (g m$^{-2}$) as a measure of species-level productivity within each plot of each year. The summed biomass for all species per plot each year was determined as grassland community-level productivity for each year. In forests, we used annual basal area increment and the accumulated basal area as estimates of species-level productivity within each plot of each year (Supplementary Methods 2). The annual basal area increment of the specific year is a measure of annual growth rate and is therefore comparable to annual aboveground biomass in grasslands. Given the longevity of trees, the accumulated basal area can also be used as a measure of cumulative productivity as forests develop[4], in which mortality was therefore taken into account. The summed annual basal area increment and accumulated basal area for all species per plot were calculated to represent the community-level productivity or cumulative productivity at a given year in experimental forest communities.

## Species overyielding and community-level overyielding

The term 'overyielding' has been used in the literature in different ways. Here, based on the aboveground productivity of each species in monocultures and mixtures, we calculated species-level response to species richness as the log response ratio (*lnRR*) of a species' observed productivity in mixtures divided by its expected value based on monoculture productivity[75]:

$$\text{Species lnRR} = \ln\left(\frac{O_i}{E_i}\right) = \ln\left(\frac{O_i}{M_i \times p_i}\right) = \ln(O_i) - \ln(M_i \times p_i) \quad (1)$$

where $O_i$ is the observed productivity of species $i$ in the mixture, $E_i$ is the expected productivity of species $i$ in the mixture, $M_i$ is the monoculture productivity of species $i$, and $p_i$ is the seeded or planted proportion of species $i$ in the mixture. The species lnRR employed in this study shares the same interpretation as the proportional deviation of species $i$'s productivity in the mixtures from its expected value[76] ($D_i = (O_i - E_i)/E_i$) and the classical definition of the relative yield of a species in the mixture ($RY_i = O_i/E_i$)[77,78]. A positive species lnRR indicates species overyielding (i.e., a species exhibits higher productivity in the mixture than expected based on its monocultural productivity and its sown or planted proportion in the mixture; equivalent to $D_i > 0$ and $RY_i/p_i > 1$), whereas negative lnRR indicates species underyielding (i.e., a species exhibits lower productivity in the mixture than expected based on its monocultural productivity; equivalent to $D_i < 0$ and $RY_i/p_i < 1$). Species lnRR approaches infinity when monoculture productivity is near zero or negative infinity when the productivity of a species in the mixture is near zero (establishment failure). Consequently, we excluded data points that were above the 99.5% percentile and lower than the 0.5% percentile[4,11,47].

We calculated community-level response to species richness as the lnRR of productivity for all species in a mixture divided by the mean productivity of monocultures of all the component species:

$$\text{Community lnRR} - \ln\left(\frac{\sum O_i}{\sum E_i}\right) = \ln\left(\frac{\sum O_i}{\sum M_i P_i}\right) = \ln\left(\sum O_i\right) - \ln\left(\sum M_i P_i\right)$$

$$(2)$$

The community lnRR employed in this study has equivalent properties to the proportional deviation of the observed total productivity ($D_T$) from its expected productivity from its corresponding monocultures ($D_T = (\Sigma O_i - \Sigma E_i)/\Sigma E_i$). Here, a positive community *lnRR* (equivalent to a positive $D_T$ and a net biodiversity effect in the additive partitioning method[5]) indicates community overyielding (i.e., the observed productivity in a mixture is higher than expected from monocultures of the constituent species), whereas a negative value (equivalent to a negative $D_T$ and a net biodiversity effect in the additive partitioning method) indicate community underyielding (i.e., the observed productivity in a mixture is lower than expected from monocultures of the constituent species). Note that the community lnRR in this study is not equivalent to a relative yield total (RYT) in the relative yields framework[77,78] which is a sum of the relative yields of each species in making its calculations. The community overyielding (or underyielding) metric we used in this study is often used to broadly define positive biodiversity effect on community productivity[18,34,35,42,57,65,75], and is different from the original definition using RYT framework from De Wit's work in the 1950s[77]. Moreover, some studies have argued that only niche complementarity and positive species interactions can generate the original concept of community overyielding[7,47,76]–which differs from the definition we used.

We also used the additive partitioning method of Loreau and Hector[5] which is also consistent with the relative yield framework[76] and which defines an overall net biodiversity effect and divides it into two classes of mechanisms: a complementarity effect and a selection effect that quantifies the covariance between species performance in monocultures and mixtures (Supplementary Methods 3).

## The contribution of species-level overyielding to community overyielding

We detected the overyielding communities (as those with a positive community-level *lnRR*), and calculated the proportion of overyielding species (with a positive species *lnRR*) in each overyielding community (PO) in each year, as the number of species with a positive *lnRR* divided by the richness of that sown or planted overyielding community. The proportion values were then divided into three groups: less than half of all species overyielded ($0 < PO < 0.5$), more than half of all species overyielded ($0.5 \leq PO < 1$), or all constituent species overyielded ($PO = 1$) in overyielding communities. Subsequently, we calculated the proportional contribution of the single highest overyielding species to community overyielding ($CO_{max}$), which was the deviation of the observed productivity of the greatest overyielding species in the mixture from its expected value divided by the deviation of the observed total productivity of the mixture from the expected total productivity:

$$CO_{max} = (O_{i(\text{lnRR max})} - M_{i(\text{lnRR max})}p_{i(\text{lnRR max})})/(\Sigma O_i - \Sigma M_i p_i) \quad (3)$$

$CO_{max}$ can be larger than 1 when overyielding by some species overcompensates for underyielding by other species, whereas $CO_{max}$ values between 0 and 1 occurs when all species overyield in overyielding communities. This calculation is not equivalent to the relative yield framework, but offers a simple way to assess whether the higher productivity in the mixture compared to the monoculture can be dominated by those greatest overyielding species. Note that the proportional contribution of all constituent species to the community-level overyielding should add up to 1.

## Statistical analysis

We used linear mixed-effect models to assess how species overyielding in mixtures was related to plant functional traits and how it changed over time, which is the interspecific trait-dependent species-level overyielding changes over time. Each model included plant traits (traits PC1, traits PC2), experimental year, plant species richness (SR), traits × SR interaction, traits × year interaction, and SR × year interaction as fixed factors. SR was log2-transformed, plant traits were

log10-transformed, and the experimental year was natural logarithm-transformed to meet the assumptions of linear mixed-effect models. In addition, random effect terms were integrated into each model, including a random intercept for each species in each unique plot (i.e., SP_PlotID), a random intercept for each site, and random intercepts and slopes for SR, traits, and year per site, which indirectly reflects the effects of the yearly climatic conditions within a given site. Furthermore, we accounted for the temporal autocorrelation due to repeated measurements of each species in plots over time by using a first-order autoregressive covariance structure with a temporal covariate in the residuals.

We also constructed alternative models by including the climate conditions (mean annual temperature (MAT) and precipitation (MAP) as well as the standardized precipitation-evapotranspiration index (SPEI)) of each corresponding year for each experimental site as a covariate in the models (Supplementary Table 7). The climate data MAT and MAP were downloaded from ERA5Land[79] (monthly averaged data), and SPEI were accessed from the global SPEI dataset[80]. We did not find significant influence of climate variables on the species-level overyielding in grassland experiments. In forest experiments, despite the effects of climate covariates on species overyielding, trait-dependent temporal shifts in species overyielding remained as expected. The models including year per site as a random factor had lower Akaike information criterion (AIC) values than the alternative models with climate conditions of each corresponding year as a covariate. As quantifying the effects of climate conditions on species overyielding was beyond our aim, we kept the models including year per site as a random factor as the final models.

We tested the reliability of using imputed traits in linear mixed-effect models by comparing the results using datasets with or without imputed traits, and found that the results of both approaches were similar (Supplementary Figs. 12–14). In addition, many grassland experiments collected data for less than four years. Therefore, we tested whether trait-dependent temporal shifts in species overyielding were consistent between the grassland experiments with more or less than four years of data. We found that the temporal trends of both datasets (i.e., all experiments ($n = 39$) versus dataset in which experiments that only had three or four years of data were excluded (i.e., only 6 experiments left)) yielded similar results (Supplementary Figs. 12 and 15). Previous studies have reported that the importance of different species to community overyielding can change by the second year of establishment[29]. Hence we reported results for the grasslands with whole datasets.

For the linear mixed-effect models testing the effects of plant species richness on community-level productivity and community overyielding (as well as complementarity and selection effects) over time, the fixed factors included SR, year, and SR × year interaction, random factors included random intercept for PlotID, a random intercept for site, random intercepts and slopes for SR and year per site (see details in Supplementary Methods 4). The linear mixed effect models were fitted with the 'lme' function in the *nlme* package[81], and model assumptions were checked by visually examining residual plots for homogeneity of variance and quantile-quantile plots for normality of residuals. All analyses were performed in R 4.2.1[82].

### Reporting summary
Further information on research design is available in the Nature Portfolio Reporting Summary linked to this article.

## Data availability
The information for each experiment, interannual climate data, and species-level trait data used in this study are archived in Figshare (https://doi.org/10.6084/m9.figshare.25148690). The public trait data can be obtained from the TRY Plant Trait Database (https://www.try-db.org/TryWeb/), the Global Root Trait (GRooT) database (https://

groot-database.github.io/GRooT/), the Botanical Information and Ecology (BIEN) Network (https://bien.nceas.ucsb.edu/bien/), and AusTraits database (https://austraits.org/). Unpublished productivity data of grassland and tree biodiversity experiments will be available from the corresponding author upon request. The unpublished productivity data of the tree diversity experiments within the TreeDivNet could be reached by contacting the network coordinators (https://treedivnet.ugent.be).

## Code availability
R code of the linear mixed-effects models is provided in the Supplementary Methods.

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

## Acknowledgements

This study was supported by the State Key Program of the National Natural Science Foundation of China (no. 32030068 to E.Y.). We recognize the invaluable contributions of all the researchers who initially developed the ideas and hypotheses, and who designed the BEF experiments in this study, maintained them, and ensured their regular measurements. We thank the platform provided by the TreeDivNet (https://www.treedivnet.ugent.be). We thank the Synthesis Centre of the German Centre for Integrative Biodiversity Research (iDiv) Halle-Jena-Leipzig (DFG FZT 118). The Jena and MyDiv experiments in this study were supported by the German Research Foundation (Jena Experiment: FOR 456, FOR 1451, FOR 5000; MyDiv: DFG–FZT; Ei 862/29-1). The Cedar Creek biodiversity experiments were supported by the NSF Cedar Creek Long-Term Ecological Research grant (DEB: 1831944) and the NSF ASCEND Biology Integration Institute (DBI:2021898). The Lanta_drt and Lanta_nutr experiments (led by V. L. and J.D.) were supported by the Czech Academy of Sciences (23-07533S, 24-11954S, RVO 67985939). The Wageningen Biodiversity Experiment was partly supported by a grant from the Dutch Organisation for Scientific Research (NWO) within the framework of the Biodiversity Program. The BIOTREE experiment has been established by the Max-Planck-Institute for Biogeochemistry Jena, Germany, and received funding through the Chair of Geobotany, Faculty of Biology, University of Freiburg, and a grant by the German Research Foundation (no. 439223434 to M.S.-L.; for data collection in 2019). We thank Prof. Dr. E. D. Schulze for initiating and supporting this project; and D. Mackensen and K. Hahner (Federal Forestry Office Thüringer Wald – Bundesforstamt Thüringer Wald, Bad Salzungen) for their support and BIOTREE site maintenance. The Kreinitz experiment is a cooperative research project funded by the Helmholtz-Centre for Environmental Research—UFZ. The EFForTS-project was funded by the Deutsche Forschungsgemeinschaft (DFG, German Research Foundation) – project ID 192626868 – SFB 990. The B-Tree set-up was funded by the University of Natural Resources and Life Sciences, Vienna. B-tree was supported by the BiodivERsA3 and BiodivClim ERA-Net COFOUND program (Dr. Forest, MixForChange), and with the funding organization Österreichischer Wissenschaftsfonds [grant numbers I 4372-B, I 50086-B]. BR and DLG were supported by the EU Horizon project Excellentia [no. 101087262]. We thank the Sardinian Forest Authority "Forestas" for hosting the IDENT-Macomer experiment and contributing to its establishment and management. IDENT-Freiburg has been financially supported by the University of Freiburg, including a grant to M.S.-L. from the "Innovationsfonds Forschung". IDENT sites in North-America were supported by the National Research and Engineering Council of Canada. Biodiversi-TREE has been supported by the Smithsonian Institution and the US National Science Foundation (NSF DEB 2044406 & Macrosystems 2106014). The Ridgefield was instigated thanks to the award of an Australian Research Council Laureate Fellowship (FL0992007) to R.J.H. The Sardinilla experiment benefited from the support of the National Research and Engineering Council of Canada, a Canada Research Chair (Tier 1) to C.Potvin and from the Smithsonian Tropical Research Institute. We thank D. Binkley, A. Weigelt and W. Weisser for contributing data, and H.Y.H. Chen, I. Ibáñez, D. He, and K. Zhu for helping with data analysis, and thank T.F. Au for editorial comments. We also thank Takehiro Sasaki and the other reviewers for their comments on the improvement of the manuscript.

## Author contributions

L. Zheng, E.Y., Y.H., K.E.B., and N.R.G.-R. conceived and developed the idea. L. Zheng, N.R.G.-R., and D.C. assembled the data. L. Zheng performed the analyses with contributions from K.E.B., N.R.G.-R., E.Y., Y.H., and P.B.R. The initial manuscript was prepared by L. Zheng, K.E.B., N.R.G.-R., Y.H., and E.Y. All authors including K.V., M.S.-L., N.E., N.B., J.B., H.B., J.C.-B., J.D., H.A., M.V.F., O.F., S.F., D.I.F., G.G., T.G., J.H., P.H., A.H., B.H., D.H., K.B.H., B.I., H.J., J.K., H.K., V.L., J.L., S. Mereu, C.M., F.M., M.M., S.Müller, B.M., C.A.N., A.P., W.C.P., J.D.P., J.A.P., G.B.P., M.P.P., D.P., H.W.P., Q.P., C.Potvin, J.Q., B.R., D.L.G., J.V.R., R.J.S., A.S., L.S., J.U., L.J.W., B.J.W., B.Y., L. Zhang, Z.Z., Y.Y., H.S., A.E., B.S., M.F., M.M.K., C. Palmborg and D.T. contributed to data collection and/or editing and improving several manuscript versions.

## Competing interests

The authors declare no competing interests.

## Additional information

Liting Zheng [1,2] ✉, Kathryn E. Barry [3], Nathaly R. Guerrero-Ramírez[4,5,6], Dylan Craven[7,8], Peter B. Reich [2,9,10], Kris Verheyen[11], Michael Scherer-Lorenzen [12], Nico Eisenhauer[13,14], Nadia Barsoum[15], Jürgen Bauhus [16], Helge Bruelheide [13,17], Jeannine Cavender-Bares [18], Jiri Dolezal [19,20], Harald Auge [13,21], Marina V. Fagundes[22], Olga Ferlian [13,14], Sebastian Fiedler [23], David I. Forrester [24], Gislene Ganade [22], Tobias Gebauer[12,25], Josephine Haase[12,26], Peter Hajek [12], Andy Hector [27], Bruno Hérault [28,29], Dirk Hölscher [6,30], Kristin B. Hulvey[31], Bambang Irawan[32,33], Hervé Jactel[34], Julia Koricheva [35], Holger Kreft[4,6], Vojtech Lanta[19,20], Jan Leps [19,36], Simone Mereu[37,38,39], Christian Messier [40,41], Florencia Montagnini[42], Martin Mörsdorf[12,43], Sandra Müller [12], Bart Muys[44], Charles A. Nock [12,45], Alain Paquette [40], William C. Parker[46], John D. Parker[47], John A. Parrotta [48], Gustavo B. Paterno [4], Michael P. Perring [49,50], Daniel Piotto[51], H. Wayne Polley[52], Quentin Ponette [53], Catherine Potvin[54], Julius Quosh[13,14], Boris Rewald [55,56], Douglas L. Godbold [55,56], Jasper van Ruijven[57,58], Rachel J. Standish [59], Artur Stefanski[9], Leti Sundawati[60], Jon Urgoiti[40], Laura J. Williams [9,10], Brian J. Wilsey[61], Baiyu Yang[1], Li Zhang[1], Zhao Zhao[1], Yongchuan Yang [62], Hans Sandén[55], Anne Ebeling[63], Bernhard Schmid [64], Markus Fischer [65], Martyna M. Kotowska [66], Cecilia Palmborg[67], David Tilman[18,68], Enrong Yan [1,69] ✉ & Yann Hautier [3]

[1]Zhejiang Zhoushan Island Observation and Research Station, Zhejiang Tiantong National Forest Ecosystem Observation and Research Station, Shanghai Key Lab for Urban and Ecological Processes and Eco-Restoration, School of Ecological and Environmental Sciences, East China Normal University, Shanghai, China. [2]Institute for Global Change Biology and School for Environment and Sustainability, University of Michigan, Ann Arbor, MI, USA. [3]Ecology and Biodiversity Group, Department of Biology, Utrecht University, Utrecht, The Netherlands. [4]Biodiversity, Macroecology and Biogeography, Faculty of Forest Sciences and Forest Ecology, University of Göttingen, Göttingen, Germany. [5]Silviculture and Forest Ecology of Temperate Zones, Faculty of Forest Sciences and Forest Ecology, University of Goettingen, Göttingen, Germany. [6]Centre of Biodiversity and Sustainable Land Use, University of Göttingen, Göttingen, Germany. [7]GEMA Center for Genomics, Ecology & Environment, Universidad Mayor, Huechuraba, Santiago, Chile. [8]Data Observatory Foundation, ANID Technology Center No. DO210001, Providencia, Santiago, Chile. [9]Department of Forest Resources, University of Minnesota, Saint Paul, MN, USA. [10]Hawkesbury Institute for the Environment, Western Sydney University, Penrith, NSW, Australia. [11]Forest & Nature Lab, Department of Environment, Faculty of Bioscience Engineering, Ghent University, Melle-Gontrode, Belgium. [12]Geobotany, Faculty of Biology, University of Freiburg, Freiburg, Germany. [13]German Centre for Integrative Biodiversity Research (iDiv) Halle-Jena-Leipzig, Leipzig, Germany. [14]Institute of Biology, Leipzig University, Leipzig, Germany. [15]Centre for Ecosystems, Society and Biosecurity, Forest Research, Alice Holt Lodge, Farnham, UK. [16]Chair of Silviculture, Faculty of Environment and Natural Resources, University of Freiburg, Freiburg, Germany. [17]Institute of Biology/Geobotany and Botanical Garden, Martin Luther University Halle Wittenberg, Halle, Germany. [18]Department of Ecology, Evolution and Behavior, University of Minnesota, Saint Paul, MN, USA. [19]Department of Botany, Faculty of Science, University of South Bohemia, České Budějovice, Czech Republic. [20]Department of Functional Ecology, Institute of Botany CAS, Třeboň, Czech Republic. [21]Department of Community Ecology, Helmholtz-Centre for Environmental Research—UFZ, Halle (Saale), Germany. [22]Departamento de Ecología, Universidade Federal do Rio Grande do Norte, Natal, Brazil. [23]Department of Ecosystem Modelling, Büsgen-Institute, University of Göttingen, Göttingen, Germany. [24]CSIRO Environment, GPO Box 1700 Canberra, ACT, Australia. [25]Bioenergy Systems Department, Resource Mobilisation, German Biomass Research Center—DBFZ gGmbH, Leipzig, Germany. [26]Department of Aquatic Ecology, Eawag—Swiss Federal Institute of Aquatic Science and Technology, Dübendorf, Switzerland. [27]Department of Biology, University of Oxford, Oxford, UK. [28]CIRAD, Forêts et Sociétés, Montpellier, France. [29]Forêts et Sociétés, Univ Montpellier, CIRAD, Montpellier, France. [30]Tropical Silviculture and Forest Ecology, Faculty of Forest Sciences and Forest Ecology, University of Göttingen, Göttingen, Germany. [31]Working Lands Conservation, Multiplier, Logan, UT, USA. [32]Forestry Department, Faculty of Agriculture, University of Jambi, Jambi, Indonesia. [33]Land Use Transformation Systems Center of Excellence, University of Jambi, Jambi, Indonesia. [34]INRAE, University of Bordeaux, BIOGECO, Cestas, France. [35]Department of Biological Sciences, Royal Holloway University of London, Egham, UK. [36]Biological Research Centre, Czech Academy of Sciences, České Budějovice, Czech Republic. [37]Consiglio Nazionale delle Ricerche, Istituto per la Bioeconomia, CNR-IBE, Sassari, Italy. [38]CMCC—Centro Euro-Mediterraneo sui Cambiamenti Climatici, IAFES Division, Sassari, Italy. [39]National Biodiversity Future Center (NBFC), Piazza Marina 61 (c/o palazzo

Steri), Palermo, Italy. [40]Département des sciences biologiques, Centre for Forest Research, Université du Québec à Montréal, Montreal, QC, Canada. [41]Département des sciences naturelles, ISFORT, Université du Québec en Outaouais, Ripon, QC, Canada. [42]School of Forestry and Environmental Studies, Yale University, New Haven, CT, USA. [43]Department for Research, Biotope-, and Wildlife Management; National Park Administration Hunsrück-Hochwald, Birkenfeld, Germany. [44]Department of Earth and Environmental Sciences, KU Leuven, Leuven, Belgium. [45]Department of Renewable Resources, Faculty of Agriculture, Life and Environmental Sciences, University of Alberta, Edmonton, AB, Canada. [46]Ontario Ministry of Natural Resources and Forestry, Sault Ste. Marie, ON, Canada. [47]Smithsonian Environmental Research Center, Edgewater, MD, USA. [48]USDA Forest Service, Research & Development, Washington, DC, USA. [49]UKCEH (UK Centre for Ecology & Hydrology), Environment Centre Wales, Bangor, UK. [50]The UWA Institute of Agriculture, The University of Western Australia, Perth, WA, Australia. [51]Centro de Formação em Ciências Agroflorestais, Universidade Federal do Sul da Bahia, Itabuna, Brazil. [52]USDA, Agricultural Research Service, Temple, TX, USA. [53]Earth and Life Institute, Université Catholique de Louvain, Louvain-la-Neuve, Belgium. [54]Department of Biology, McGill University, Montréal, QC, Canada. [55]Forest Ecology, Department of Forest and Soil Sciences, University of Natural Resources and Life Sciences, Vienna, Austria. [56]Forest Ecosystem Research, Department of Forest Protection and Wildlife Management, Faculty of Forestry and Wood Technology, Mendel University in Brno, Brno, Czech Republic. [57]Plant Ecology and Nature Conservation Group, Wageningen University, Wageningen, The Netherlands. [58]Forest Ecology and Management group, Wageningen University, Wageningen, The Netherlands. [59]School of Environmental and Conservation Sciences, Murdoch University, Murdoch, WA, Australia. [60]Department of Forest Management, Faculty of Forestry and Environment, Institut Pertanian Bogor University, Bogor, Indonesia. [61]Department of Ecology, Evolution and Organismal Biology, Iowa State University, Ames, IA, USA. [62]Key Laboratory of the Three Gorges Reservoir Region's Eco-Environment, Ministry of Education, Chongqing University, Chongqing, China. [63]Institute of Ecology and Evolution, University Jena, Jena, Germany. [64]Department of Geography, University of Zurich, Zurich, Switzerland. [65]Institute of Plant Sciences, University of Bern, Bern, Switzerland. [66]Department of Plant Ecology and Ecosystems Research, University of Göttingen, Göttingen, Germany. [67]Department of Crop Production Ecology, Swedish University of Agricultural Sciences, Umeå, Sweden. [68]Bren School of Environmental Science and Management, University of California, Santa Barbara, CA, USA. [69]Institute of Eco-Chongming (IEC), Shanghai, China. ✉e-mail: litingz@umich.edu; eryan@des.ecnu.edu.cn

