## [Peer Review File · Nature Communications]

Effects of plant diversity on productivity strengthen over time due to trait-dependent shifts in species overyieldingEditorial Note: This manuscript has been previously reviewed at another journal that is not operating a transparent peer review scheme. This document only contains reviewer comments and rebuttal letters for versions considered at *Nature Communications*.

Reviewer #1 (Remarks to the Author):

Overall, I think the question “Whether the strengthening of diversity effects is caused by temporal shifts in species overyielding” is worth to test with large dataset across ecosystems, as this study did here. The manuscript is well written and the result is reasonable. However, my concern is still about the data and the models, although I appreciate the authors spent quite some effort in explaining or improving the reliability of the data compared to the last version:

1) For the productivity/overyielding data: Although the authors emphasize in this version that they actually used 39 grassland and 26 forest experiments through all the analysis, I found the available temporal data from the grassland are not exactly “long-term”. This is clearly showed in Table S1. Among the 39 grassland experiments, 31 has the data from year 1-3, 2 has the data from year 1-4, only 6 has the data longer than 8 years. Figure S12 and S15 only showed the predicted lines, the models and number of species used for these two figures are missing. To my understanding, this is a bit dangerous to use most shorter time data (79%) to predict much longer temporal pattern. The forest data sounds more rationale to predict the long-term pattern.

2) For the trait data: I understand that not each species can get all five traits, but there should be a table in the SI showing that for each trait, how many species have the measured and imputed values for the two ecosystems. That will help to quickly understand the reliability of the trait data.

3) For the linear mixed models: The random factors in most of the linear mixed models are a bit confusing. e.g. for Table 1, the ddf of Year, Traits PC1 × Year, SR × Year in the Table 1 is so large. Then I found (Table S4) they used the random intercept or slope of a few variables as random factors. I wonder why plot is missing from the random terms. This is probably the reason why the ddf is so large.

Reviewer #3 (Remarks to the Author):

<Comments for the Authors:NCOMMS-23-54034-T>

I was invited to act as the third reviewer to evaluate the revised version of the manuscript “Effects of plant diversity on productivity strengthen over time due to trait-dependent shifts in species overyielding”. Most of reviewers’ concerns on the earlier version of the manuscript were addressed in a satisfying manner. Except for minor comments below, I do not have anything else to add.

Abstract. Somewhere in the Abstract, the authors may want to mention that plant ecological strategies here were defined by functional traits.

Supplementary Table 1. For the Wageningen dataset, species richness range would be 1, 2, 4, and 8 (not 12, 4, 8).

Takehiro Sasaki

REVIEWERS' COMMENTS

Reviewer #1

Overall, I think the question “Whether the strengthening of diversity effects is caused by temporal shifts in species overyielding” is worth to test with large dataset across ecosystems, as this study did here. The manuscript is well written and the result is reasonable. However, my concern is still about the data and the models, although I appreciate the authors spent quite some effort in explaining or improving the reliability of the data compared to the last version:

1) For the productivity/overyielding data: Although the authors emphasize in this version that they actually used 39 grassland and 26 forest experiments through all the analysis, I found the available temporal data from the grassland are not exactly “long-term”. This is clearly showed in Table S1. Among the 39 grassland experiments, 31 has the data from year 1-3, 2 has the data from year 1-4, only 6 has the data longer than 8 years. Figure S12 and S15 only showed the predicted lines, the models and number of species used for these two figures are missing. To my understanding, this is a bit dangerous to use most shorter time data (79%) to predict much longer temporal pattern. The forest data sounds more rationale to predict the long-term pattern.

Response: We thank the reviewer for pointing this out. We acknowledge that one limitation of this study is the imbalance in dataset length among grassland experiments, which is a source of uncertainty in the analyses. Therefore, we performed further analyses to test whether trait-dependent temporal shifts in species overyielding were consistent between the grassland experiments with greater than or fewer than four years of data. We found that the temporal trends of the restricted longer-term dataset (i.e., only 6 experiments left) yielded similar results (Supplementary Figs. 12 and 15) as those presented above using the full datasets (i.e., all 39 experiments). Our results indicate that the variation in number of years of data across the experiments and the unavailability of longer-term datasets, do not compromise the conclusions. In the new revised version, we added the text about the findings from the dataset restricted to longer-term datasets in the Results (L311-321), and also discussed the implications of the imbalance in dataset length for interpreting and extrapolating the findings in the discussion (L460-466).

In the supplementary figures for the additional analyses, we used the same linear mixed effect models testing species overyielding over time but just restricted the datasets. We showed both the predicted lines and the test statistic (F , df and P values) of the interactive term of traits \times year, which mainly determined the predicted pattern of trait-dependent species overyielding over time. Now we have added the relevant information to the legend of each supplementary figure.

2) For the trait data: I understand that not each species can get all five traits, but there should be a table in the SI showing that for each trait, how many species have the measured and imputed values for the two ecosystems. That will help to quickly understand the reliability of the trait data.

Response: Thank you for the suggestion. We have added the number of species with measured and imputed values for each trait in Supplementary Table 3 and cited it in the main text (L530-531).

3) For the linear mixed models: The random factors in most of the linear mixed models are a bit confusing. e.g. for Table 1, the ddf of Year, Traits PC1 \times Year, SR \times Year in the Table 1 is so large. Then I found (Table S4) they used the random intercept or slope of a few variables as random factors. I wonder why plot is missing from the random terms. This is probably the reason why the ddf is so large.

Response: In the species-level model, we used the unique labels of each species within each unique plot (i.e., SP_PlotID) to indicate that species are nested in a plot (i.e., the random term in the model $\sim 1|SP_PlotID$ is equivalent to species $\sim 1|Plot/SP$). We used unique labels in the random structure because the random effects and the temporal autocorrelation structure (repeated measurements of each species in plots over time) have to match in the *lme()* function. In addition to the random terms, the very large *dfs* in the models are also related to the sample size when the observation is each species in each plot.

Reviewer #3

I was invited to act as the third reviewer to evaluate the revised version of the manuscript “Effects of plant diversity on productivity strengthen over time due to trait-dependent shifts in species overyielding”. Most of reviewers’ concerns on the earlier version of the manuscript were addressed in a satisfying manner. Except for minor comments below, I do not have anything else to add.

Abstract. Somewhere in the Abstract, the authors may want to mention that plant ecological strategies here were defined by functional traits.

Response: We have revised this sentence as “These temporal trends of species-level overyielding are shaped by plant ecological strategies, which can be quantitatively delimited by functional traits” (L124).

Supplementary Table 1. For the Wageningen dataset, species richness range would be 1, 2, 4, and 8 (not 12, 4, 8).

Response: We have made the correction as suggested by Reviewer 3.